# Social-Inverse: Inverse Decision-making of Social Contagion Management with Task Migrations

**Guangmo Tong**

Department of Computer and Information Sciences

University of Delaware

`amotong@udel.edu`

## Abstract

Considering two decision-making tasks $A$ and $B$, each of which wishes to compute an effective *decision $Y$* for a given *query $X$*, can we solve task $B$ by using query-decision pairs $(X, Y)$ of $A$ without knowing the latent decision-making model? Such problems, called *inverse decision-making with task migrations*, are of interest in that the complex and stochastic nature of real-world applications often prevents the agent from completely knowing the underlying system. In this paper, we introduce such a new problem with formal formulations and present a generic framework for addressing decision-making tasks in social contagion management. On the theory side, we present a generalization analysis for justifying the learning performance of our framework. In empirical studies, we perform a sanity check and compare the presented method with other possible learning-based and graph-based methods. We have acquired promising experimental results, confirming for the first time that it is possible to solve one decision-making task by using the solutions associated with another one.

## 1 Introduction

**Social contagion management.** Social contagion, in its most general sense, describes the diffusion process of one or more information cascades spreading between a set of atomic entities through the underlying network [1, 2, 3, 4]. Prototypical applications of social contagion management include deploying advertising campaigns to maximize brand awareness [5, 6], broadcasting debunking information to minimize the negative impact of online misinformation [7, 8, 9], HIV prevention for homeless youth [10, 11], and the prevention of youth obesity [12, 13]. In these applications, a central problem is to launch new information cascades in response to certain input queries, with the goal of optimizing the agents' objectives [14, 15]. In principle, most of these tasks fall into either *diffusion enhancement*, which seeks to maximize the influence of the to-be-generated cascade (e.g., marketing campaign [5, 16, 17] and public service announcement [12, 18, 19]), or *diffusion containment*, which aims to generate positive cascades to minimize the spread of negative cascades (e.g., misinformation [20, 21, 22] and violence-promoting messages [23, 24]).

**Inverse decision-making with task migrations.** Traditional research on social contagion management often adopts classic operational diffusion models with known parameters, and focuses on algorithmic development in overcoming the NP-hardness [25, 17, 26, 27, 28]. However, real-world contagions are often very complicated, and therefore, perfectly knowing the diffusion model is less realistic [29, 30, 31, 32, 33]. When presented with management tasks defined over unknown diffusion models, one can adopt the learn-and-optimize approach in which modeling methodologies and optimization schemes are designed separately; in such methods, the main issue is that the learning process is guided by model accuracy but not by the optimization effect [34, 35], suggesting that

36th Conference on Neural Information Processing Systems (NeurIPS 2022).

the endeavors dedicated to model construction are neither necessary nor sufficient for successfully handling the downstream optimization problems. This motivates us to explore unified frameworks that can shape the learning pipeline towards effective approximations. Recently, it has been shown that for contagion management tasks like diffusion containment, it is possible to produce high-quality decisions for future queries by using query-decision pairs from the same management task without learning the diffusion model [36]. Such findings point out an interesting and fundamental question: with a fixed latent diffusion model, can we solve a target management task by using query-decision pairs from a different management task? This is of interest because the agents often simultaneously deal with several management tasks while it is less likely that they always have the proper empirical evidence concerning the target task. For example, a network manager may need to work on a rumor blocking task, but they only have historical data collected in solving viral marketing tasks. We call such a setting as *inverse decision-making with task migrations*.

**Contribution.** This paper presents a formal formulation of inverse decision making where the target task we wish to solve is different from the source task that generates samples, with a particular focus on social contagion management tasks. Our main contribution is a generic framework, called *Social-Inverse*, for handling migrations between tasks of diffusion enhancement and diffusion containment. For Social-Inverse, we present theoretical analysis to obtain insights regarding how different contagion management tasks can be subtly correlated in order for samples from one task to help the optimization of another task. In empirical studies, we have observed encouraging results indicating that our method indeed works the way it is supposed to. Our main observations suggest that the task migrations are practically manageable to a satisfactory extent in many cases. In addition, we also explore the situations where the discrepancy between the target task and the source task is inherently essential, thereby making the samples from the source task less useful.

**Roadmap.** In Sec. 2, we first provide preliminaries regarding social contagion models, and then discuss how to formalize the considered problem. The proposed method together with its theoretical analysis is presented in Sec. 3. In Sec. 4, we present our empirical studies. We close our paper with a discussion on limitations and future works (Sec. 5). The technical proofs, source code, pre-train models, data, and full experimental analysis can be found in the supplementary material. The data and source code is maintained online[1].

## 2 Preliminaries

### 2.1 Stochastic diffusion model

A social network is given by a directed graph $G = (V, E)$, with $V$ and $E$ respectively denoting the user set and the edge set. In modeling the contagion process, let us assume that there are $L \in \mathbb{Z}$ information cascades $\{C_i\}_{i=1}^L$, each of which is associated with a seed set $S_i \subseteq V$. Without loss of generality, we assume that $S_i \cap S_j = \emptyset$ for $i \neq j$. A diffusion model $\mathcal{M}$ is governed by two sets of configurations: each node $u \in V$ is associated with a distribution $\mathcal{N}_u$ over $2^{N_u^-}$, where $N_u^-$ is the set of the in-neighbors of $u$; each edge $(u, v) \in E$ is associated with a distribution $\mathcal{T}_{(u,v)}$ over $(0, +\infty)$ denoting the transmission time. During the diffusion process, a node can be inactive or $C_i$-active if activated by cascade $C_i$. Given the seed sets, the diffusion process unfolds as follows:

- **Initialization**: Each node $u$ samples a subset $A_u \subseteq N_u^-$ following $\mathcal{N}_u$, and each edge $(u, v)$ samples a real number $t_{(u,v)} \geq 0$ following $\mathcal{T}_{(u,v)}$.
- **Time** 0: The nodes in $S_i$ become $C_i$-active at time 0, and other nodes are inactive.
- **Time** $t$: When a node $u$ becomes $C_i$-active at time $t$, for each inactive node $v$ such that $u$ is in $A_v$, $v$ will be activated by $u$ and become $C_i$-active at time $t + t_{(u,v)}$. Each node will be activated by the first in-neighbor attempting to activate them and never deactivated. When a node $v$ is activated by two or more in-neighbors at the same time, $v$ will be activated by the cascade with the smallest index.

**Remark 1.** The considered model is in general highly expressive because $\mathcal{N}_u$ and $\mathcal{T}_{(u,v)}$ can be flexibly designed. For sample, it subsumes the classic independent cascade model [25] by making $\mathcal{N}_u$ sample each in-neighbor independently. When there is only one or two cascades, the above model generalizes a few popular diffusion models, including discrete-time independent cascade model [25],

---

[1]`https://github.com/cdslabamotong/social_inverse`

discrete-time linear threshold model [25], continuous-time independent cascade model [37], and independent multi-cascade model [38, 36].

**Definition 1** (**Realization**). Notice that the initialization phase essentially samples a weighted subgraph, and the diffusion process becomes deterministic after the initialization phase. For an abstraction, we call each of such weighted subgraph a *realization*, and use $\mathcal{R}_G$ to denote the space of weighted subgraphs of $G$. With the concept of realization, we may abstract a concrete stochastic diffusion model $\mathcal{M}$ as a collection of density functions, i.e., $\mathcal{M} = \{\mathcal{N}_u : u \in V\} \cup \{\mathcal{T}_{(u,v)} : (u,v) \in E\}$. Slightly abusing the notation, we also use $\mathcal{M} : \mathcal{R}_G \to [0,1]$ to denote the distribution over $\mathcal{R}_G$ induced by the density functions specified by $\mathcal{M}$. On top of a diffusion model $\mathcal{M}$, the distribution of the diffusion outcome depends on the seed sets of the cascades. An example for illustrating the diffusion process is given in Appendix A.

## 2.2 Social contagion management tasks

In this paper, we focus on the following two classes of social contagion management tasks.

**Problem 1** (**Diffusion Enhancement (DE)**). Given a diffusion model $\mathcal{M}$ and a set of target users $X \subseteq V$, we consider the single-cascade case and let $f_{\mathcal{M}}^{\mathrm{DE}}(X, Y)$ be the expected number of users in $X$ who are activated by a cascade from seed set $Y \subseteq V$. We would like to find a seed set with at most $k \in \mathbb{Z}$ nodes such that the total influence on $X$ can be maximized, i.e.,

$$\underset{Y \subseteq V, |Y| \leq k}{\arg\max} \; f_{\mathcal{M}}^{\mathrm{DE}}(X, Y). \tag{1}$$

**Problem 2** (**Diffusion Containment (DC)**). Given a diffusion model $\mathcal{M}$, we now consider the situation of competitive diffusion where there is a negative cascade $C_1$ with a seed set $X \subseteq V$ and a positive cascade $C_2$ with a seed set $Y \subseteq V$. Let $f_{\mathcal{M}}^{\mathrm{DC}}(X, Y)$ be the expected number of users who are *not* activated by the negative cascade. Given the diffusion model $\mathcal{M}$ and the seed set $X$ of the negative cascade, we would like to find a seed set $Y$ for the positive cascade with at most $k \in \mathbb{Z}$ nodes such that the impact of the negative cascade can be maximally limited, i.e.,

$$\underset{Y \subseteq V, |Y| \leq k}{\arg\max} \; f_{\mathcal{M}}^{\mathrm{DC}}(X, Y). \tag{2}$$

Contagion management tasks like Problems 1 and 2 might be viewed as decision-making problems aiming to infer an effective decision $Y$ in response to a query $X$. In such a sense, we may abstract such problems in the following way:

**Problem 3** (**Abstract Contagion Management Tasks**). Given a diffusion model $\mathcal{M}$, an abstract management task $T$ is specified by an objective function $f_{\mathcal{M}}^T(X, Y) : 2^V \times 2^V \to \mathbb{R}$, a candidate space $\mathcal{X}_T \subseteq 2^V$ of the queries, and a candidate space $\mathcal{Y}_T \subseteq 2^V$ of the decisions, where we wish to compute

$$Y_{\mathcal{M}, T, X} := \underset{Y \in \mathcal{Y}_T}{\arg\max} \; f_{\mathcal{M}}^T(X, Y) \tag{3}$$

for each input query $X \in \mathcal{X}_T$. We assume that $\mathcal{X}_T$ and $\mathcal{Y}_T$ are matroids over $V$, subsuming common constraints such as the cardinality constraint or $k$-partition [39]. Since such optimization problems are often NP-hard, their approximate solutions are frequently used, and we denote by $Y_{\mathcal{M}, T, X}^{\alpha}$ an $\alpha$-approximation to Equation 3.

**Definition 2** (**Linearity over Kernel Functions**). In addressing the above management tasks, it is worth noting that the objective function is calculated over the possible diffusion outcomes, which are determined in the initialization phase. Specifically, denoting by $f_r^T(X, Y)$ the objective value projected to a single realization $r \in \mathcal{R}_G$, the objective function can be expressed as

$$f_{\mathcal{M}}^T(X, Y) = \int_{r \in \mathcal{R}_G} \mathcal{M}(r) \cdot f_r^T(X, Y) dr. \tag{4}$$

The function $f_r^T(X, Y)$ is called *kernel function*, in the sense that it transforms set structures into real numbers. For example, $f_r^{\mathrm{DE}}(X, Y)$ denotes the number of users in $X$ who are activated by a cascade generated from $Y$ in a single realization $r$; $f_r^{\mathrm{DC}}(X, Y)$ denotes the number of users in $G$ who are not activated by the negative cascade (generated from $X$) in realization $r$ when the positive cascade spreads from $Y$.

## 2.3 Inverse decision-making of contagion management with task migrations

Supposing that the diffusion model $\mathcal{M}$ is given, DE and DC are purely combinatorial optimization problems, which have been extensively studied [25, 20]. In the case that the diffusion model is unknown, inverse decision-making is a potential solution, which seeks to solve contagion management tasks by directly learning from query-decision pairs [40]. In particular, with respect to a certain management task $f_{\mathcal{M}}^T$ associated with an unknown diffusion model $\mathcal{M}$, the agent receives a collection of pairs $(X_i, Y_i)$ where $Y_i$ is the optimal/suboptimal solution to maximizing $f_{\mathcal{M}}^T(X_i, Y)$. Such empirical evidence can be mathematically characterized as

$$S_{\mathcal{M},T,m}^{\overline{\alpha}} = \left\{ (X_i, Y_{\mathcal{M},T,X_i}^{\overline{\alpha}}) : f_{\mathcal{M}}^T(X_i, Y_{\mathcal{M},T,X_i}^{\overline{\alpha}}) \geq \overline{\alpha} \cdot \max_{Y \in \mathcal{Y}_T} f_{\mathcal{M}}^T(X_i, Y) \right\}_{i=1}^{m} \tag{5}$$

where $\overline{\alpha} \in (0, 1]$ is introduced to measure the optimality of the sample decisions. For the purpose of theoretical analysis, the ratio $\overline{\alpha}$ may be interpreted as the best approximation ratio that can be achieved by a polynomial-time algorithm under common complexity assumptions (e.g., $NP \neq P$). For DE and DC, we have the best ratio as $1 - 1/e$ due to the well-known fact that their objective functions are submodular [25, 20]. Leveraging such empirical evidence, we wish to solve the same or a different management task for future queries:

**Problem 4** (**Inverse Decision-making with Task Migrations**). Suppose that there is an underlying diffusion model $\mathcal{M}_{true}$. Consider two management tasks $T_{\circledS}$ and $T_{\circledT}$ defined in Problem 3, where $T_{\circledS}$ is the source task and $T_{\circledT}$ is the target task. With a collection $S_{\mathcal{M}_{true}, T_{\circledS}, m}^{\overline{\alpha}_{\circledS}}$ of samples concerning the source task $T_{\circledS}$ for some ratio $\overline{\alpha}_{\circledS} \in (0, 1]$, we aim to build a framework $A : \mathcal{X}_{T_{\circledT}} \to \mathcal{Y}_{T_{\circledT}}$ that can make a prediction $A(X)$ for each future query $X$ of the target task $T_{\circledT}$. Let $l$ be a loss function $l(X, \hat{Y}) : \mathcal{X}_{T_{\circledT}} \times \mathcal{Y}_{T_{\circledT}} \to [0, 1]$ that measures the desirability of $\hat{Y}$ with respect to $X$. We seek to minimize the generalization error $\mathcal{L}$ with respect to an unknown distribution $\mathcal{D}$ over $\mathcal{X}_{T_{\circledT}}$:

$$\mathcal{L}(A, \mathcal{D}, l) \coloneqq \mathbb{E}_{X \sim \mathcal{D}} \left[ l(X, A(X)) \right]. \tag{6}$$

Since $\mathcal{M}_{true}$ and $m$ are fixed, we denote $S_{\mathcal{M}_{true}, T_{\circledS}, m}^{\overline{\alpha}_{\circledS}}$ as $S_{T_{\circledS}}^{\overline{\alpha}_{\circledS}}$ for conciseness. We will focus on the case where the source task and the target task are selected from DE and DC.

**Remark 2.** In general, the above problem appears to be challenging because the query-decision pairs of one optimization problem do not necessarily shed any clues on effectively solving another optimization problem. What makes our problem tractable is that the source task and the target task share the same underlying diffusion model $\mathcal{M}_{true}$. With the hope that the query-decision pairs of the source task can identify $\mathcal{M}_{true}$ to a certain extent, we may solve the target task with statistical significance, as evidenced in experiments. In such a sense, our setting is called *inverse* as it implicitly infers the structure of the underlying model from solutions, in contrast to the forward decision-making pipeline that seeks solutions based on given models.

# 3 Social-Inverse

In this section, we present a learning framework called Social-Inverse for solving Problem 4. Our method is inspired by the classic structured prediction [41] coupled with randomized kernels [40], which may be ultimately credited to the idea of Random Kitchen Sink [42]. Social-Inverse starts by selecting an empirical distribution $\mathcal{M}_{em}$ over $\mathcal{R}_G$ and a hyperparameter $K \in \mathbb{Z}$, and then proceeds with the following steps:

- **Hypothesis design.** Sample $K$ iid realizations $R_K = \{r_1, ..., r_K\} \subseteq \mathcal{R}_G$ following $\mathcal{M}_{em}$, and obtain the hypothesis space $\mathcal{F}_{R_K} \coloneqq \{H_{R_K, \mathbf{w}}^{\circledT}(X, Y) : \mathbf{w} = (w_1, ..., w_K) \in \mathbb{R}^K\}$ where $H_{R_K, \mathbf{w}}^{\circledT}$ is the affine combination of $f_r^{T_{\circledT}}$ over the realizations in $R_K$:

$$H_{R_K, \mathbf{w}}^{\circledT}(X, Y) \coloneqq \sum_{i=1}^{K} w_i \cdot f_{r_i}^{T_{\circledT}}(X, Y). \tag{7}$$

- **Training.** Compute a prior vector $\widetilde{\mathbf{w}}$ using the training set $S_{T_{\circledS}}^{\overline{\alpha}_{\circledS}}$ (Sec. 3.2), and sample the final parameter $\overline{\mathbf{w}}$ from an isotropic Gaussian $Q(\gamma \cdot \widetilde{\mathbf{w}}, \mathcal{I})$ with a mean of $\widetilde{\mathbf{w}}$ scaled by $\gamma$. The selection of $\gamma$ will be discussed in Sec. 3.1.

- **Inference.** Given a future query $X \in \mathcal{X}_{T_{①}}$ of the target task $T_{①}$, the prediction is made by solving the inference problem over the hypothesis in $\mathcal{F}_{R_K}$ associated with final weight $\overline{\mathbf{w}}$:

$$\arg\max_{Y \in \mathcal{Y}_{T_{①}}} H^{①}_{R_K,\overline{\mathbf{w}}}(X,Y). \tag{8}$$

It is often NP-hard to solve the above inference problem in optimal, and therefore, we assume that an $\widetilde{\alpha}_{①}$-approximation to Equation 8 – denoted by $Y^{\widetilde{\alpha}_{①}}_{R_K,\overline{\mathbf{w}},X}$ – is employed for some $\widetilde{\alpha}_{①} \in (0,1]$. Notice that the ratio $\widetilde{\alpha}_{①}$ herein represents the inference hardness, while the ratio $\overline{\alpha}_{⑤}$ associated with the training set measures the hardness of the source task.

In completing the above procedure, it is left to determine a) the prior distribution $\mathcal{M}_{em}$, b) the hyperparameter $K$, c) the scale factor $\gamma$, d) the training method for computing the prior vector $\widetilde{\mathbf{w}}$, and e) the inference algorithm for computing $Y^{\widetilde{\alpha}_{①}}_{R_K,\overline{\mathbf{w}},X}$. In what follows, we will first discuss how they may influence the generalization performance in theory, and then present methods for their selections. For the convenience of reading, the notations are summarized in Table 2 in Appendix B.

### 3.1 Generalization analysis

For Social-Inverse, given the fact the generation of $\overline{\mathbf{w}}$ is randomized, the generalization error is further expressed as

$$\mathcal{L}(\text{Social-Inverse}, \mathcal{D}, l) := \mathbb{E}_{X \sim \mathcal{D}, \overline{\mathbf{w}} \sim Q}\left[l(X, Y^{\widetilde{\alpha}_{①}}_{R_K,\overline{\mathbf{w}},X})\right]. \tag{9}$$

In deriving an upper bound with respect to the prior vector $\widetilde{\mathbf{w}}$, let us notice that the empirical risk is given by $l(X_i, Y^{\widetilde{\alpha}_{①}}_{R_K,\overline{\mathbf{w}},X_i})$, which is randomized by $\overline{\mathbf{w}} \sim Q(\gamma \cdot \widetilde{\mathbf{w}}, \mathcal{I})$. Thus, the prediction $Y^{\widetilde{\alpha}_{①}}_{R_K,\overline{\mathbf{w}},X_i}$ associated with a training input $X_i$ is most likely one of those centered around $Y^{\widetilde{\alpha}_{①}}_{R_K,\widetilde{\mathbf{w}},X_i}$, and we will measure such concentration by their difference in terms of a fraction of the empirical risk associated with $\widetilde{\mathbf{w}}$. More specifically, controlled by a hyperparameter $\beta \in (0, \widetilde{\alpha}_{①})$, for an input query $X_i$, the potential predictions are those within the margin:

$$\mathrm{I}^{①}_{R_K,\widetilde{\mathbf{w}},X_i,\beta} := \tag{10}$$
$$\left\{Y \in \mathcal{Y}_{①} : \widetilde{\alpha}_{①} \cdot H^{①}_{R_K,\widetilde{\mathbf{w}}}(X_i, Y^{\widetilde{\alpha}_{①}}_{R_K,\widetilde{\mathbf{w}},X_i}) - H^{①}_{R_K,\widetilde{\mathbf{w}}}(X_i, Y) \leq \beta \cdot H^{①}_{R_K,\widetilde{\mathbf{w}}}(X_i, Y^{\widetilde{\alpha}_{①}}_{R_K,\widetilde{\mathbf{w}},X_i})\right\}.$$

The empirical risk is therefore given via the above margin:

$$\mathcal{L}_{em}(\widetilde{\mathbf{w}}, R_K, S^{\overline{\alpha}_{⑤}}_{T_{⑤}}, l) := \frac{1}{m}\sum_{i=1}^{m} \max_{Y \in \mathcal{Y}_{①}} l(X_i, Y) \cdot \mathbb{1}_{\mathrm{I}^{①}_{R_K,\widetilde{\mathbf{w}},X_i,\beta}}(Y), \tag{11}$$

where $\mathbb{1}_S(x) \in \{0,1\}$ is the indicator function: $\mathbb{1}_S(x) = 1 \iff x \in S$. With the above progressions, we have the following result concerning the generalization error.

**Theorem 1.** *For each $\widetilde{\mathbf{w}} = (\widetilde{\mathbf{w}}_1, ..., \widetilde{\mathbf{w}}_K)$, $R_K \subseteq \mathcal{R}_G$, $\beta \in (0, \widetilde{\alpha}_{①})$, and $\delta > 0$, with probability at least $1 - \delta$, we have*

$$\mathcal{L}(\text{Social-Inverse}, \mathcal{D}, l) \leq \mathcal{L}_{em}(\widetilde{\mathbf{w}}, R_K, S^{\overline{\alpha}_{⑤}}_{T_{⑤}}, l) + \frac{\|\widetilde{\mathbf{w}}\|^2}{m} + \sqrt{\frac{\gamma^2\|\widetilde{\mathbf{w}}\|^2/2 + \ln(m/\delta)}{2(m-1)}}$$

*provided that*

$$\gamma = \frac{\widetilde{\alpha}_{①}^2 + 1}{\min_p |\widetilde{\mathbf{w}}_p| \cdot \beta \cdot \widetilde{\alpha}_{①}}\sqrt{2\ln\frac{2mK}{\|\widetilde{\mathbf{w}}\|^2}} \tag{12}$$

The proof follows from the standard analysis of the PAC-Bayesian framework [43] coupled with the approximate inference [44] based on a multiplicative margin [40]; the extra caution we need to handle is that our margin (Equation 10) is parameterized by $\beta$. Notice that when $\beta$ decreases, the regularization term becomes larger, while the margin set $\mathrm{I}^{①}_{R_K,\widetilde{\mathbf{w}},X_i,\beta}$ becomes smaller – implying a low empirical risk (Equation 11). In this regard, Theorem 1 presents an intuitive trade-off between the estimation error and the approximation error controlled by $\beta$.

Having seen the result for a general loss, we now seek to understand the best possible generalization performance in terms of the approximation loss $l_{\text{approx}}$:

$$l_{\text{approx}}(X, \hat{Y}) \coloneqq 1 - \frac{f_{\mathcal{M}_{true}}^{T_{\oplus}}(X, \hat{Y})}{\max_{Y \in \mathcal{Y}_{\oplus}} f_{\mathcal{M}_{true}}^{T_{\oplus}}(X, Y)} \in [0, 1]. \tag{13}$$

Such questions essentially explore the realizability of the hypothesis space $\mathcal{F}_{R_K}$, which is determined by the empirical distribution $\mathcal{M}_{em}$ and the number of random realizations used to construct $\mathcal{F}_{R_K}$. We will see shortly how these factors may impact the generalization performance. By Theorem 1, when infinite samples are available, the empirical risk approaches to

$$\mathbb{E}_{X \sim \mathcal{D}} \left[ \max_{Y \in \mathcal{Y}_{T_{\oplus}}} l_{\text{approx}}(X, Y) \cdot \mathbb{1}_{\mathrm{I}_{R_K, \widetilde{\mathbf{w}}, X, \beta}^{\oplus}}(Y) \right]. \tag{14}$$

The next result provides an analytical relationship between the complexity of the hypothesis space and the best possible generalization performance in terms of $l_{\text{approx}}$.

**Theorem 2.** *Let $\Delta \coloneqq \sup_r \frac{\mathcal{M}_{true}(r)}{\mathcal{M}_{em}(r)} \cdot \frac{\max f_r^{T_{\oplus}}(X,Y)}{\min f_r^{T_{\oplus}}(X,Y)}$ measure the divergence between $\mathcal{M}_{true}$ and $\mathcal{M}_{em}$ scaled by the range of the kernel function. For each $\epsilon > 0, \delta_1 > 0, \delta_2 > 0$, and $\mathcal{M}_{em}$, when $K$ is $O(\frac{\Delta^2}{\epsilon^2 \cdot \delta_2^2}(\ln |\mathcal{Y}_{T_{\oplus}}| + \ln \frac{1}{\delta_1}))$, with probability at least $1 - \delta_1$ over the selection of $R_K$, there exists a desired weight $\widetilde{\mathbf{w}}$ such that*

$$\Pr_{X \sim \mathcal{D}} \left[ \max_{Y \in \mathcal{Y}_{T_{\oplus}}} l_{\text{approx}}(X, Y) \cdot \mathbb{1}_{\mathrm{I}_{R_K, \widetilde{\mathbf{w}}, X, \beta}^{\oplus}}(Y) \leq 1 - \frac{\widetilde{\alpha}_{\oplus} \cdot (\widetilde{\alpha}_{\oplus} - \beta) \cdot (1 - \epsilon)}{(1 + \epsilon)} \right] \geq 1 - \delta_2. \tag{15}$$

**Remark 3.** The above result has the implication that the best possible ratio in generalization is essentially bounded by $O(\widetilde{\alpha}_{\oplus} \cdot (\widetilde{\alpha}_{\oplus} - \beta))$. On the other hand, one can easily see that the target task (Equation 3) and the inference problem (Equation 8) suffer the same approximation hardness, and therefore, one would not wish for a true approximation error that is better than $\widetilde{\alpha}_{\oplus}$; in this regard, the result in Theorem 2 is not very loose.

The results in this section demonstrate how the selections of $K, \mathcal{M}_{em}, \widetilde{\mathbf{w}}$, and $\beta$ may affect the generalization performance in theory. Since the true model $\mathcal{M}_{true}$ is unknown, the prior distribution $\mathcal{M}_{em}$ can be selected to be uniform or Gaussian distribution. $K$ and $\beta$ can be taken as hyperparameters determining the model complexity. In addition, since the true loss $l_{\text{approx}}$ is not accessible, one can take general loss functions. Given the fact that we are concerned with set structures rather than real numbers, we employ the zero-one loss, which is adopted also for the convenience of optimization. Therefore, it remains to figure out how to compute the prior vector $\widetilde{\mathbf{w}}$ from training samples as well as how to solve the inference problem (Equation 8), which will be discussed in the next part.

## 3.2 Training method

In computing the prior vector $\widetilde{\mathbf{w}}$, the main challenge caused by the task migration is that the target task on which we performance inference is different from the source task that generates training samples. Theorem 1 suggests that, ignoring the low-order terms, one may find the prior vector by minimizing the regularized empirical risk $\mathcal{L}_{em}(\widetilde{\mathbf{w}}, R_K, S_{T_{\oplus}}^{\overline{\alpha}_{\ominus}}, l) + \frac{\|\widetilde{\mathbf{w}}\|^2}{m}$. Directly minimizing such a quantity would be notoriously hard because the optimization problem is bilevel: optimizing over $\widetilde{\mathbf{w}}$ involves the term $Y_{R_K, \widetilde{\mathbf{w}}, X_i}^{\widetilde{\alpha}_{\oplus}}$ which is obtained by solving another optimization problem depending on $\widetilde{\mathbf{w}}$ (Equations 10 and 11). Notably, since $H_{R_K, \widetilde{\mathbf{w}}}^{\oplus}(X_i, Y_{R_K, \widetilde{\mathbf{w}}, X_i}^{\widetilde{\alpha}_{\oplus}})$ is lower bounded by $\widetilde{\alpha}_{\oplus} \cdot H_{R_K, \widetilde{\mathbf{w}}}^{\oplus}(X_i, Y_{\mathcal{M}_{true}, T_{\oplus}, X_i})$, replacing $Y_{R_K, \widetilde{\mathbf{w}}, X_i}^{\widetilde{\alpha}_{\oplus}}$ with $Y_{\mathcal{M}_{true}, T_{\oplus}, X_i}$ would allow for us to optimize an upper bound of the empirical risk. Seeking a large-margin formulation, this amounts to solving the following mathematical program under the zero-one loss [41, 45]:

$$\min \quad \|\mathbf{w}\|^2 + C \sum_{i=1}^{m} \xi_i / m$$

$$\text{s.t.} \quad \widetilde{\alpha}_{\oplus}(\widetilde{\alpha}_{\oplus} - \beta) \cdot H_{R_K, \mathbf{w}}^{\oplus}(X_i, Y_{\mathcal{M}_{true}, T_{\oplus}, X_i}) - H_{R_K, \mathbf{w}}^{\oplus}(X_i, Y) \geq \xi_i, \ \forall i \in [m], \ \forall Y \in \mathcal{Y}_{\oplus}$$

$$\mathbf{w} \geq 0 \tag{16}$$

where $\xi_i$ is the slack variable and $C$ is a hyperparameter [46]. However, our dataset concerns only about the source task $T_{\circledS}$ without informing $Y_{\mathcal{M}_{true}, T_{\circledT}, X_i}$ or its approximation. In order to see where we could feed the training samples into the training process, let us notice that the constraints in Equation 16 have an intuitive meaning: with respect to the target the task $T_{\circledT}$, a desired weight $\widetilde{\mathbf{w}}$ should lead to a score function $H_{R_K, \widetilde{\mathbf{w}}}^{\circledT}$ that can assign highest scores to the optimal solutions $Y_{\mathcal{M}_{true}, T_{\circledT}, X_i}$. Similar arguments also apply to the source task $T_{\circledS}$, as the weight $\widetilde{\mathbf{w}}$ implicitly estimates the true model $\mathcal{M}_{true}$, which is independent of the management tasks. This enables us to reformulate the optimization problem with respect to the source task $T_{\circledS}$ by using the following constraints:

$$\widetilde{\alpha}_{\circledT}(\widetilde{\alpha}_{\circledT} - \beta) \cdot H_{R_K, \mathbf{w}}^{\circledS}(X_i, Y_{\mathcal{M}_{true}, T_{\circledS}, X_i}^{\overline{\alpha}_{\circledS}}) - H_{R_K, \mathbf{w}}^{\circledS}(X_i, Y) \geq \xi_i, \ \forall i \in [m], \ \forall Y \in \mathcal{Y}_{\circledT} \quad (17)$$

where $H_{R_K, \mathbf{w}}^{\circledS}(X, Y) \coloneqq \sum_{i=1}^{K} w_i \cdot f_{r_i}^{T_{\circledS}}(X, Y)$ is the score function corresponding to the source task $T_{\circledS}$. As desired, pairs of $(X_i, Y_{\mathcal{M}_{true}, T_{\circledS}, X_i}^{\overline{\alpha}_{\circledS}})$ are the exactly the information we have in the training data $S_{T_{\circledS}}^{\overline{\alpha}_{\circledS}}$. One remaining issue is that the acquired program (Equation 17) has an exponential number of constrains [45], which can be reduced to linear (in sample size) if the following optimization problem can be solved for each $\mathbf{w}$ and $X_i$:

$$\max_{Y \in \mathcal{Y}_{\circledS}} H_{R_K, \mathbf{w}}^{\circledS}(X_i, Y). \quad (18)$$

Provided that the above problem can be addressed, the entire program can be solved by several classic algorithms, such as the cutting plane algorithm [47] and the online subgradient algorithm [48]. Therefore, in completing the entire framework, it remains to solve Equations 8 and 18. For tasks of DE and DC, we delightedly have the following results.

**Theorem 3.** *When $T_{\circledS}$ and $T_{\circledT}$ are selected from DE and DC, Equations 8 and 18 are both NP-hard to solve in optimal, but both can be approximated within a ratio of $1 - 1/e$ in polynomial time.*

A concrete example of using Social-Inverse to solve Problem 4 is provided in Appendix C.

## 4 Empirical studies

Although some theoretical properties of our framework can be justified (Sec. 3.1), it remains open whether or not the proposed method is practically effective, especially given the fact that no prior work has attempted to solve one optimization problem by using the solutions to another one. In this section, we present our empirical studies.

### 4.1 Experimental settings

We herein present the key logic of our experimental settings and provide details in Appendix E.1.

**The latent model $\mathcal{M}_{true}$ and samples (Appendix E.1.1).** To generate a latent diffusion model $\mathcal{M}_{true}$, we first determine the graph structure and then fix the distributions $\mathcal{N}_u$ and $\mathcal{T}_{(u,v)}$ by generating random parameters. We adopt four graphs: a Kronecker graph [49], an Erdős-Rényi graph [50], a Higgs graph [51], and a Hep graph [52]). Given the underlying diffusion model $\mathcal{M}_{true}$, for each of DE and DC, we generate a pool of query-decision pairs $(X_i, Y_i)$ for training and testing, where $X_i$ is selected randomly from $V$ and $Y_i$ is the approximate solution associated with $X_i$ (Theorem 3). As for Problem 4, there are four possible target-source pairs: DE-DE, DC-DE, DE-DC, and DC-DC.

**Social-Inverse (Appendix E.1.2).** With $K$ and $\beta$ being hyperparameters, to set up Social-Inverse, we need to specify the empirical distribution $\mathcal{M}_{em}$. We construct the empirical distribution by building three diffusion models $\mathcal{M}_q$ with $q \in \{0.1, 0.5, 1\}$, where a smaller $q$ implies that $\mathcal{M}_q$ is closer to $\mathcal{M}_{true}$. In addition, we construct an empirical distribution $\mathcal{M}_\infty$ which is totally random and not close to $\mathcal{M}_{true}$ anywhere. For each empirical distribution, we generate a pool of realizations.

**Competitors (Appendix E.1.3).** Given the fact that Problem 4 may be treated as a supervised learning problem with the ignorance of task migration, we have implemented Naive Bayes (NB) [53], graph neural networks (GNN) [54], and a deep set prediction network (DSPN) [55]. In addition, we consider the High-Degree (HD) method, which is a popular heuristic believing that selecting the high-degree users as the seed nodes can decently solve DE and DC. A random (Random) method is also used as the baseline.

Table 1: **Results on Kronecker (Kro) and Erdős-Rényi (ER).** Each cell shows the mean of performance ratio with std. For Social-Inverse under each empirical distribution, the table shows the results with $K \in \{15, 30, 60\}$ and $\beta = 1$. The training size is 270, and the testing size is 540.

| | | | K = 15 | K = 30 | K = 60 | | K = 15 | K = 30 | K = 60 |
|---|---|---|---|---|---|---|---|---|---|
| **Kro** | DC \| DE | $\mathcal{M}_\infty$ | 0.541 (0.014) | 0.567 (0.016) | 0.584 (0.006) | $\mathcal{M}_1$ | 0.748 (0.022) | 0.794 (0.016) | 0.811 (0.011) |
| | | $\mathcal{M}_{0.5}$ | 0.767 (0.028) | 0.837 (0.004) | 0.851 (0.004) | $\mathcal{M}_{0.1}$ | 0.770 (0.025) | 0.850 (0.018) | 0.853 (0.040) |
| | DE | **NB**: 0.64 (0.01) | **GNN**: 0.51 (0.05) | **DSPN**: 0.46 (0.14) | | **HD**: 0.59 (0.01) | **Random**: 0.15 (0.01) | | |
| | DE \| DC | $\mathcal{M}_\infty$ | 0.846 (0.005) | 0.913 (0.005) | 0.985 (0.014) | $\mathcal{M}_1$ | 0.796 (0.056) | 0.845 (0.040) | 0.899 (0.030) |
| | | $\mathcal{M}_{0.5}$ | 0.850 (0.036) | 0.937 (0.058) | 1.021 (0.040) | $\mathcal{M}_{0.1}$ | 0.862 (0.013) | 0.953 (0.018) | 1.041 (0.026) |
| | DC | **NB**: 0.88 (0.01) | **GNN**: 0.76 (0.05) | **DSPN**: 0.57 (0.14) | | **HD**: 0.89 (0.01) | **Random**: 0.26 (0.01) | | |
| **ER** | DC \| DE | $\mathcal{M}_\infty$ | 0.541 (0.010) | 0.585 (0.018) | 0.577 (0.007) | $\mathcal{M}_1$ | 0.744 (0.005) | 0.752 (0.003) | 0.754 (0.003) |
| | | $\mathcal{M}_{0.5}$ | 0.825 (0.005) | 0.830 (0.005) | 0.830 (0.005) | $\mathcal{M}_{0.1}$ | 0.829 (0.002) | 0.833 (0.004) | 0.833 (0.004) |
| | DE | **NB**: 0.78 (0.01) | **GNN**: 0.05 (0.02) | **DSPN**: 0.06 (0.03) | | **HD**: 0.32 (0.01) | **Random**: 0.06 (0.01) | | |
| | DE \| DC | $\mathcal{M}_\infty$ | 0.526 (0.020) | 0.677 (0.016) | 0.750 (0.010) | $\mathcal{M}_1$ | 0.773 (0.016) | 0.796 (0.010) | 0.800 (0.009) |
| | | $\mathcal{M}_{0.5}$ | 0.879 (0.002) | 0.900 (0.002) | 0.892 (0.002) | $\mathcal{M}_{0.1}$ | 0.886 (0.005) | 0.902 (0.003) | 0.895 (0.003) |
| | DC | **NB**: 0.04 (0.01) | **GNN**: 0.05 (0.02) | **DSPN**: 0.04 (0.01) | | **HD**: 0.09 (0.01) | **Random**: 0.04 (0.01) | | |

**Training, testing, and evaluation (Appendix E.1.4)** The testing size is 540, and the training size $m$ is selected from $\{90, 270, 1350\}$. Given the training size and the testing size, the samples are randomly selected from the pool we generate; similarly, given $K$, the realizations used in Social-Inverse are also randomly selected from the pool we generate. For each method, the entire process is repeated five times, and we report the average performance ratio together with the standard deviation. The performance ratio is computed by comparing the predictions with the decisions in testing samples; larger is better.

## 4.2 Main observations

The main results on the Kronecker graph and the Erdős-Rényi graph are provided in Table 1. According to Table 1, it is clear that Social-Inverse performs better when $K$ becomes larger or when the discrepancy between $\mathcal{M}_{em}$ and $\mathcal{M}_{true}$ becomes smaller (i.e., $q$ is small), which suggests that Social-Inverse indeed works the way it is supposed to. In addition, while all the methods are non-trivially better than Random, one can also observe that Social-Inverse easily outperforms other methods by an evident margin as long as sufficient realizations are provided. We also see that learning-based methods do not perform well in many cases; this is not very surprising because the effectiveness of learning-based methods hinges on the assumption that different tasks share similar decisions for the same query, which however may not be the case especially on the Erdős-Rényi graph. Furthermore, Social-Inverse appears to be more robust than other methods in terms of standard deviation. Finally, the performance of standard learning methods (e.g., NB and GNN) are sensitive to graph structures; they are relatively good on the Kronecker graph but less satisfactory on the Erdős-Rényi graph, while Social-Inverse is consistently effective on all datasets.

## 4.3 An in-depth investigation on task migration

Notably, the effectiveness of Social-Inverse depends not only on the training samples $S_{T_\oplus}^{\overline{\alpha}_\oplus}$ (for tuning the weight $\overline{\mathbf{w}}$) but also on the expressiveness of the hypothesis space (determined by $\mathcal{M}_{em}$ and $K$). Therefore, with solely the results in Table 1, we are not ready to conclude that samples of DE (resp., DC) are really useful for solving DC (resp., DE). In fact, when $\mathcal{M}_{em}$ is identical to $\mathcal{M}_{true}$, no samples are needed because setting $\overline{\mathbf{w}} = 1$ can allow for us to perfectly recover the best decisions as long as $K$ is sufficiently large. As a result, the usefulness of the samples should be assessed by examining how much they can help in delivering a high-quality $\overline{\mathbf{w}}$. To this end, for each testing query, we report the quality of two predictions made based, respectively, on the initial weight (before optimization) and on the final weight (after optimization).

Such results for DC-DE on the Kronecker graph are provided in Figure 1. As seen from Figure 1b, the efficacy of DC samples in solving DE is statistically significant under $\mathcal{M}_{0.1}$, which might be the first

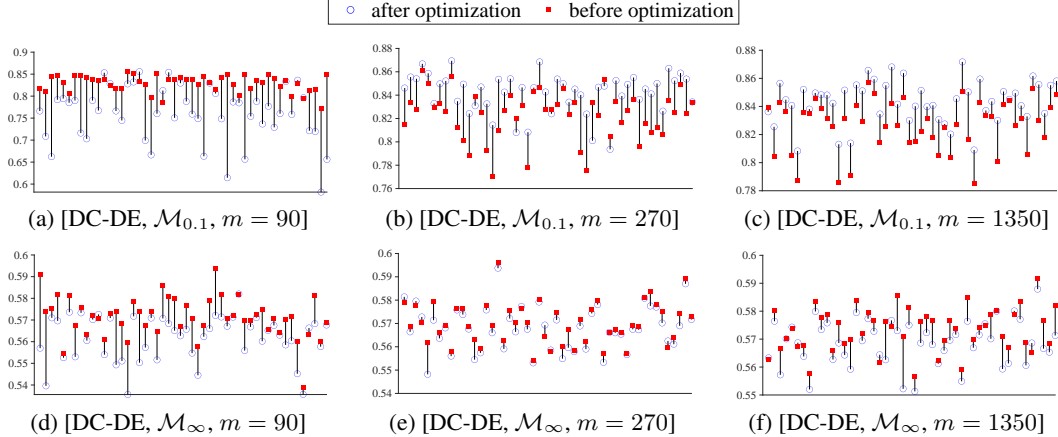

Figure 1: Each subgraph shows the results of 50 testing queries on the Kronecker graph with $K = 30$. For each query, we report the quality (i.e. performance ratio) of the predictions made based on the initial weight (before optimization) and final weight (after optimization).

piece of experimental evidence confirming that it is indeed possible to solve one decision-making task by using the query-decision pairs of another one. In addition, according to Figures 1a, 1b, and 1c, the efficacy of the samples is limited when the sample size is too small, and it does not increase too much after sufficient samples are provided. On the other hand, with Figure 1e, we also see that the samples of DC are not very useful when the empirical distribution (e.g., $\mathcal{M}_{\infty}$) deviates too much from the true model, and in such a case, providing more samples can even cause performance degeneration (Figure 1f), which is an interesting observation calling for further investigations.

The full experimental study can be found in Appendix E.2, E.3 and E.4, including the results on Higgs and Hep, results with more realizations, results of DE-DE and DC-DC, results of GNN under different seeds, a discussion on the impact of $\beta$, and a discussion of training efficiency.

## 5    Further discussion

We close our paper with a discussion on the limitations and future directions of the presented work, with the related works being discussed in Appendix F.

**Future directions.** Following the discussion on Figure 1, one immediate direction is to systemically investigate the necessary or sufficient condition in which the task migrations between DE and DC are manageable. In addition, settings in Problem 4 can be carried over to other contagion management tasks beyond DE and DC, such as effector detection [56] and diffusion-based community detection [57]. Finally, in its most general sense, the problem of inverse decision-making with task migrations can be conceptually defined over any two stochastic combinatorial optimization problems [40] sharing the same underlying model (e.g., graph distribution). For instance, considering the stochastic shortest path problem [58] and the minimum Steiner tree problem [59], with the information showing the shortest paths between some pairs of nodes, can we infer the minimum Steiner tree of a certain group of nodes with respect to the same graph distribution? Such problems are interesting and fundamental to many complex decision-making applications [60, 61].

**Limitations.** While our method offers promising performance in experiments, it is possible that deep architectures can be designed in a sophisticated manner so as to achieve improved results. In addition, while we believe that similar observations also hold for graphs that are not considered in our experiment, more experimental studies are required to support the universal superiority of Social-Inverse. In another issue, the assumption that the training set contains approximation solutions is the minimal one for the purpose of theoretical analysis, but in practice, such guarantees may never be known. Therefore, experimenting with samples of heuristic query-decision pairs is needed to further justify the practical utility of our method. Finally, we have not experimented with graphs of extreme scales (e.g., over 1M nodes) due to the limit in memory. We wish to explore the above issues in future work.

## Acknowledgments and Disclosure of Funding

We thank the reviewers for their time and insightful comments. This work is supported in part by a) National Science Foundation under Award IIS-2144285 and b) the University of Delaware.

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
