# OpenReview forum: "Social-Inverse: Inverse Decision-making of Social Contagion Management with Task Migrations"
_NeurIPS.cc/2022/Conference — NeurIPS 2022 Accept_

### Official Review · Reviewer_HDGT · 2022-07-05

**Rating:** 5
**Confidence:** 3
**Soundness:** 4 excellent
**Presentation:** 2 fair
**Contribution:** 3 good

**Summary:**

The paper presents a framework for task migration in the context of contagion management in social networks, in which, a target decision-making task is solved using the historical data collected from another task. The authors define a generic problem formulation that covers the tasks of diffusion enhancement as well as diffusion containment, and further derive a reformulated optimization problem that can be coupled with several classic algorithms to provide an approximated solution in a polynomial time.

**Questions:**

- How the concepts of "task migration" relate to "transfer learning"?

- It is not completely clear what the authors mean by learning-based methods. Isn't your approach a learning framework?



**Limitations:**

No discussion is provided in that regard. However, given the nature of the work, which is related to the social network analysis, a discussion about societal impact of the approach would be interesting.

**Strengths And Weaknesses:**

Strengths:

+ Originality: The paper develops a novel framework for task migration between two classes of social management tasks with a generic problem definition and (re-)formulation of the objectives.

+ Quality: The authors theoretically analyze the generalization ability of their approach across tasks and show that the generalization error is bounded. Subsequently, they identify potential conditions on the design choices that can affect the generalizability, and thus, some aspects/parameters are chosen accordingly. In addition, they conduct considerable empirical study evaluating the performance of their method in various setups and compared to several baselines.

Weaknesses:

- Significance: The proposed approach is not well located within the literature, and the discussion of the related work seems to be limited. Additionally, in terms of approaches, it is mentioned that the ideas are closely related to structure prediction methods (refs. 40-42), but the  connection to those work is not clarified; what are the similarities/differences with those methods? in what sense the framework is novel compared to them? Moreover, the generalization of the approach to other domains and problems and the applicability of the work on real network data is not clear. As a result, the outcomes only impact a narrow scope which leads to limited significance.

- Clarity: The paper is well written, but it is a bit hard to follow. The writeup could be improved in terms of structure and choice of content, Some details about the background information and algorithmic characteristic can be expanded.

---

> ### Author Response · Authors · 2022-07-31
> **Author response**
>
> We thank the reviewer for their time and constructive comments. In what follows, we will attempt to address some of the concerns.
>
> - $\textbf{Related work.}$ We share the same feeling with the reviewer that the topic studied in this paper may not be perfectly classified into any of the existing research areas, and our work is at the intersection of social network analysis, combinatorial optimization, and learning theory. We follow the high-level idea of using a score function to quantify the decision qualities, which is inspired by the classic structured prediction, but our entire framework (as well as the proofs) cannot be immediately acquired from any of the existing works. We will re-organize Section F in the supplementary materials to better explain the connections between different components.
>
> - $\textbf{Novelty.}$ Traditional methods for data-driven decision making typically follow a two-step framework: learn a decision model from data, and then solve the optimization problem based on the learned model. For example, one can first learn the diffusion model, and then solve the management tasks [a]. In contrast to the two-step framework, we seek to build decision-making pipelines using query-decision pairs for at least two reasons. First, learning the decision model can be another challenging task, which may be unnecessary if one can access query-decision pairs. Second, learning high-quality decisions directly from observed query-decision pairs is not only of theoretical interest but can also open new research avenues in designing decision-making diagrams. Following this branch, the main novelty of this paper is to introduce the concept of task migration, together with the proposed learning framework and the presented analysis. In addition, we are excited about the acquired experimental results, confirming for the first time that task migrations are indeed manageable. We will rephrase the introduction to further clarify the novelty based on the reviewer’s comments.
>
>
> - $\textbf{Generalization to other domains.}$ We agree with the reviewer that it is not super clear whether the proposed approach can be applied to other application sectors. It is our aspiration to explore the issue of task migrations in more general settings or other applications, and this paper attempts to offer a primitive formal study on a specific application. In this regard, we are motivated by other applications where the idea of query-decision regression (as well as the task migration issue) may be of interest: one is the direct perception framework [b] that seeks to compute a safe driving (decision) based on the sensor inputs (query), without learning the perception modules; another application is the DARPA Subterranean Challenge [c], where the underground surface (analogous to the diffusion model under our context) is hidden and therefore query-decision regression can be a potential decision-making diagram.
>
> - $\textbf{Representation.}$ We thank the reviewer for pointing out the representation issues, and we are happy to add more details and intuitions to make the paper more accessible to a wider group of audience.
>
>
> - $\textbf{Task migration vs. transfer learning.}$ To our knowledge, task migration happens when one wishes to solve one optimization problem using query-decision pairs associated with another optimization problem, while transfer learning in general deals with the case when training data and testing data are generated from different distributions/domains [d].
>
> - $\textbf{Miscellaneous.}$ The reviewer is correct that our method is also a learning-based method. Our experiments involve our method (Social-Inverse), other learning-based methods (NB, GNN, and DSPN), a graph-based method (High-Degree), and a random baseline. We will clarify this point in our revisions.
>
> [a] Du, Nan, Le Song, Manuel Gomez Rodriguez, and Hongyuan Zha. "Scalable influence estimation in continuous-time diffusion networks." Advances in neural information processing systems 26 (2013).
>
> [b] Chen, Chenyi, Ari Seff, Alain Kornhauser, and Jianxiong Xiao. "Deepdriving: Learning affordance for direct perception in autonomous driving." In Proceedings of the IEEE international conference on computer vision, pp. 2722-2730. 2015.
>
> [c] Rouček, Tomáš, Martin Pecka, Petr Čížek, Tomáš Petříček, Jan Bayer, Vojtěch Šalanský, Daniel Heřt et al. "Darpa subterranean challenge: Multi-robotic exploration of underground environments." In International Conference on Modelling and Simulation for Autonomous Systems, pp. 274-290. Springer, Cham, 2019.
>
> [d] Weiss, Karl, Taghi M. Khoshgoftaar, and DingDing Wang. "A survey of transfer learning." Journal of Big data 3, no. 1 (2016): 1-40.

---

### Official Review · Reviewer_8vwD · 2022-07-10

**Rating:** 7
**Confidence:** 4
**Soundness:** 3 good
**Presentation:** 4 excellent
**Contribution:** 4 excellent

**Summary:**

This paper studies an inverse decision-making problem for task migrations on contagion management. Specifically, it investigates how prior decision-making on diffusion containment can be migrated to diffusion enhancement; and vice versa. It provides a theoretical analysis of how different diffusion management tasks can be correlated such that samples from one task can be useful for another task. It performs empirical analysis on four graphs, comparing with several benchmarks (used for supervised learning) to evaluate the performance.

**Questions:**


(1) In remark 1, the paper mentioned that the model generalizes popular diffusion models, including the linear threshold model, which requires a percentage or number of neighbors to adopt before the users adopt. However, in lines 79—80 on P2, the paper mentions that the node will only be activated by "first in-attempting neighbors" or activated by the cascade with the smallest index. These two scenarios both conflict with the linear threshold model. Why can't a node be activated by two or more cascades? In reality, a user only observes his/her neighbor's decisions, rather than which cascade resulted in the neighbors' decision.

(2) L88–89, why is the subgraph weighted? From the description in Line 71—72, I think the subgraph is attributed instead of weighted?

(3) There are two decision-making problems in the paper, one is the decision-making of users, and the other is the decision-making of the central planner (I believe referred to as agent by the paper). Given that both are important concepts yet are entirely different, I suggest distinguishing the two concepts to avoid confusing the readers, e.g., user decision-marking vs. planner decision-making).

(4) \beta is an important parameter that controls the trade-off between estimation and approximation errors. How can it be tuned? It seems that the experiment section directly uses beta = 1. If it cannot be systematically tuned, it might also help to show the robustness of the method w.r.t. different beta

(5) Regarding the experiment, I think it will help add a benchmark, which assumes either a linear threshold model or an independent cascade. The parameters in these models can be learned based on different realizations. That is, since there is rich research on the diffusion model in sociology, it seems sensible to use them directly in such a migration task.

(6) I wonder how realistic in practice do we assume the platform has information about all DC but no DE, or all DE but no DC? It seems more reasonable to observe a mix of both. Can you provide some insights on how the method performs (especially compared with the benchmarks) if this is the case?

(7) Any insights on why the benchmarks perform this badly on ER?

(8) Minor: There is a typo on line 84, "For sample" —> "For example"


**Strengths And Weaknesses:**

Strengths:
I find the idea of task migration on contagion management novel and important (although I have some concerns about the detailed setup (see weakness 1). The paper provides rigorous theoretical analysis and shows empirical effectiveness on four datasets.

Weaknesses:
(1) I think the setup that the platform only observes one type of task (either diffusion containment or enhancement) does not seem realistic. It is more likely that the platform will observe a combination of both. I think this is a crucial question: How realistic is the task migration problem proposed in this paper?

(2)  Given that many models have been proposed in the diffusion literature (also cited by the paper), designing benchmarks w.r.t. based on these models is important to show the effectiveness of the methods, rather than other ML methods that do not know the underlying diffusion process.

---

> ### Author Response · Authors · 2022-07-30
> **Author response**
>
> First of all, we thank the reviewer for their time and comprehensive comments.
>
> - $\textbf{Data type.}$ It is indeed more realistic that the platform may observe a mix of two or more types of tasks instead of observing only one type. In the presence of multi-type observations, the proposed framework cannot be directly applied, and our feeling is that new algorithms and analysis must be designed, which seems technically non-trivial. For example, the constraints in the structured SVM need to be updated to accommodate different task types, and as a result, the cutting plane algorithm has to be re-designed to prioritize the constraints. We thank the reviewer for pointing out this issue, which we believe is an important future direction.
>
> - $\textbf{Benchmarks on specific models.}$ It is true that the benchmark based on specific models would be a nice one, and it is also a very interesting idea to first learn an IC or LT model based on different realizations and then consider task migrations. We thank the reviewer for these inspiring comments. We were not able to find simple benchmarks because the existing works rely on the assumption that the diffusion model is known. On the other hand, considering that the materials in this paper are already heavy, we have limited our experiments to synthetic models. We hope to implement the reviewer's ideas in the following works.
>
> - $\textbf{On Remark 1.}$ The classic linear threshold (LT) model proposed in [a] was designed for single-cascade diffusion, and in the single-cascade case, our model generalizes the LT model because the LT model can be equivalented described as to first sample a live-edge graph and then execute the deterministic diffusion process [a]. We will revise our paper with more clarifications. As for the multi-cascade case, we assume that a node can be activated by only one cascade in order to model the cascade competition; otherwise, influence containment might be impossible as two cascades will spread independently. In another issue, a user can sometimes observe not only the neighbor’s decision but also the associated cascade; for example, when a user likes/retweets a message with misinformation, we may say that they are activated by the misinformation cascade. However, we must admit that user behaviors in real networks are much more complicated than our modeling assumptions.
>
> - $\textbf{On beta.}$ Beta can be systematically tuned (by modifying the constraints in Equation 16), and a brief discussion can be found in the supplementary material E.3. As seen from there, the impact of beta seems limited, and the results are robust.
>
> - $\textbf{On ER graph.}$ We do not have a firm answer, but one plausible reason is that compared to other graphs, the ER graph is generated by uniformly selecting neighbors and therefore lacks salient graph structures, which makes the correlation between the query and decision hard to learn without using a good hypothesis space. Our method suffers less because our hypothesis space ensures to contain functions that can well approximate the latent objective function (Lemma 3 in the proof of Theorem 2).
>
> - $\textbf{Miscellaneous.}$ The diffusion graph is labeled with transmission distributions, and it becomes a weighted subgraph after realization sampling. We thank the reviewer for pointing out the typo as well as the suggestions regarding distinguishing two types of decision makings. We will address them in our revision.
>
> [a] Kempe, David, Jon Kleinberg, and Éva Tardos. "Maximizing the spread of influence through a social network." In Proceedings of the ninth ACM SIGKDD international conference on Knowledge discovery and data mining, pp. 137-146. 2003.

---

> > ### Comment · Reviewer_8vwD · 2022-08-09
> > **Thank you very much for your detailed response.**
> >
> > Thank you very much for your detailed response. I have no further comments and will keep my score.

---

### Official Review · Reviewer_iB7C · 2022-07-13

**Rating:** 4
**Confidence:** 3
**Ethics Flag:** Yes
**Soundness:** 2 fair
**Presentation:** 2 fair
**Contribution:** 3 good

**Summary:**

The paper considers a semi-opaque-box influence network and the cumulative impacts from information dissemination at a few seed nodes. The network is semi-opaque in the sense that we know the general connectivity of the nodes, but not the actual realization of the influence, which is sampled from an unknown distribution with support by the observed connections. To make up for the observability gap, the paper suggests to use previous examples of information diffusion, collected in the form of which targets to reach (queries) and which seeds would maximize the coverage of the targets after diffusion (decisions). It then proposes to use random samples of possible realizations to solve a max-margin (SVM) optimization problem to find out the most likely linear combination of the realizations. The paper finally uses the learned weighted combination of realizations to decide for optimal solutions in new information diffusion tasks. Theoretical and empirical studies were included.

**Questions:**

* Line 172. Why do you introduce a Gaussian distribution after you learned the optimal combinations of realizations? Doesn't this introduce pure noise?
* Line 185. Related, the generalization error considers the mean of the sample you use for inference, yet in the algorithm description, you use only one point in the sample for inference. Does this generalization error cover the full risk?
* Line 191. I do not understand the use of beta. Also, why do you analyze a binary loss when your objective contains an optimization sub-problem?
* Table 1. Can you elaborate on Naive Bayes? It seems to be the second-best in the considered baselines, just below High Degree.

**Ethics Review Area:**

["Inappropriate Potential Applications & Impact  (e.g., human rights concerns)"]

**Limitations:**

Overall, I can get some inspirations from the proposed method and empirical studies, but some key aspects in the algorithm were left unexplained, namely eta and the extra step of Gaussian sampling. The theory presentation does not meet the expectation of the NeurIPS community, perhaps because too many new concepts were introduced without good explanations.

Though the paper is purely methodological, I flagged it for ethics reviews due to some word choices, such as information containment, which may lead to a limitation of people's access to opportunities. The authors should be also be advised on the creation of an ethics discussion section.

**Strengths And Weaknesses:**

Strengths:
* Quality: The paper is nicely presented with thorough empirical studies.
* Originality: The paper solves an essentially "graph-kernel" problem using inspirations from random kitchen sink. This sounds like a reasonable idea.

Weaknesses:
* Theory: The presentation is not entirely clear and I have some questions about some unexplained terms in the key algorithm and theoretical analysis.
* Significance: We have to accept the assumption that the training data is presented as query-decision pairs, instead of the more commonly used decision-impact pairs, that is, the actual coverage caused by the seed nodes.
* Ethics: The paper is missing ethics discussions on a potentially sensitive topic.

---

> ### Author Response · Authors · 2022-07-31
> **Author response**
>
> We appreciate the reviewer for their time and comments, especially for pointing out the ethic concerns.
>
> - $\textbf{Presentation.}$ We share the same view with the reviewer that this paper comes with heavy notations and technical concepts, which is primarily because the presented analysis is at the confluence of social contagion management and machine learning theory, which are traditionally not very close to each other. We will carefully check the paper and add more explanations. In this regard, it would be very helpful if the reviewer could advise on the concepts that need more explanations.
>
> - $\textbf{On Gaussian prior.}$ The reviewer is correct that introducing the Gaussian distribution does not fundamentally change the learning outcome. Directly using the learned weights can be taken as a special case that uses the mean of a Gaussian. Following the convention of PAC-Bayesian [a], we wish to study the general case in the theoretical analysis.
>
> - $\textbf{Data type and significance.}$ Traditional methods for data-driven decision makings typically follow a two-step framework: learn a decision model from data, and then solve the optimization problem based on the learned model. For example, one can utilize samples of decision-impact pairs to first learn the diffusion model, and then solve the management tasks [b]. Taking a further step, we study the decision-making diagram using query-decision pairs with task migrations for at least two reasons: first, learning the decision model can be another challenging task, which may be unnecessary if one can access query-decision pairs; second, inferring high-quality decisions directly from observed query-decision pairs is not only of theoretical interest but can also open new research avenues in designing decision-making pipelines. This paper offers a primitive formal study on a specific application, hoping to inspire applications beyond social contagion management, for example, the direct perception framework [c] and DARPA's subterranean challenge [d].
>
>
> - $\textbf{Generalization error.}$ The generalization bound is taken with respect to the latent data distribution ($D$ in Equation 6), and it characterizes the prediction accuracy in expectation (over the entire domain). The inference for a future query is based on the model learned using the entire sample set.
>
> - $\textbf{On beta and binary loss.}$ The beta introduced in Equation 10 controls the tightness of the margin, in contrast to the existing works where the slack of the margin is constant [e]. Theorem 1 implies that a larger beta leads to a larger empirical risk but a smaller regularization term. In that sense, beta offers a means of balancing different terms, thereby making the true approximation error tunable (Theorem 2). A binary loss coupled with our margin makes the empirical error scaled by the margin difference; combined with Theorem 2, this indicates that the empirical error is in fact scaled by the approximation error, which allows us to factor the sub-optimization problem into the learning process. We thank the reviewer for pointing out the above two issues, and we will add more clarifications to the manuscript.
>
> - $\textbf{On Naïve Bayes.}$ It is true that Naïve Bayes is more effective than most of the baselines. One plausible reason is that Naïve Bayes makes less strong assumptions compared to deep architectures (DSPN and GNN), and therefore, it offers better generalization performance because of Occam's razor principle.
>
> - $\textbf{Ethic concerns.}$ The information containment we study here is inherited from the study of rumor blocking and misinformation prevention (e.g., [f, g]). It aims to stop/slow the spread of negative cascades but is not designed as a means of regulating or manipulating online speeches. Clearly, misuse of such methods can lead to consequences such as limiting people's access to opportunities. While our research focuses on modeling and theoretical analysis, we agree with the reviewer that this paper should have discussed such issues, and we are happy to do so in our revisions.
>
> [a] McAllester, David. "Simplified PAC-Bayesian margin bounds." In Learning theory and Kernel machines, 2003.
>
> [b] Du Nan et al. "Influence function learning in information diffusion networks." In ICML, 2014.
>
> [c] Chen Chenyi et al. "Deepdriving: Learning affordance for direct perception in autonomous driving." In ICCV, 2015.
>
> [d] Rouček Tomáš et al. "Darpa subterranean challenge: Multi-robotic exploration of underground environments." In MESAS, 2019.
>
> [e] Wu Yuanbin et al. "A learning error analysis for structured prediction with approximate inference." In NeurIPS, 2017.
>
> [f] Saxena Akrati et al. "Mitigating misinformation in online social network with top-k debunkers and evolving user opinions." In WWW, 2020.
>
> [g] Budak Ceren et al. "Limiting the spread of misinformation in social networks." In WWW, 2011.

---

### Official Review · Reviewer_egd8 · 2022-07-14

**Rating:** 6
**Confidence:** 1
**Soundness:** 3 good
**Presentation:** 1 poor
**Contribution:** 3 good

**Summary:**

The authors consider the problem of task migration for decision making tasks in social networks for the purpose of managing contagion events. The authors begin by formally defining a social network (and associated social contagions) as a directed graph of users, where a set of seed users initiate a social contagion, with contagions spreading through edges at a rate defined by a distribution on each edge.

The authors then define four different contagion management problems, the last of which is the task migration problem. In this setting, we have two contagion management tasks (source and target) on the same social network and have samples from the source task. The goal is to minimize a loss with respect to the target task using those source task samples.

The authors propose an algorithm for this task migration problem which they dub social-inverse. They bound the generalization error of social inverse and compare their algorithm in simulated experiments to supervised learning algorithms and a heuristic that selects high-degree users as seed nodes.

**Questions:**

How closely do the assumptions of this work mirror real-world practices in social contagion management?
For example, the introduction states that "For example, a network manager may need to work on a rumor blocking task, but they only have historical data collected in solving viral marketing tasks." are there any existing cases where such a network manager would employ any methods that are remotely similar to social-inverse instead of just proceeding based on intuition/industry best practices for their interventions?

**Limitations:**

Limitations were well addressed by the author.

**Strengths And Weaknesses:**

This paper addresses an important problem (task migration) in a very general setting that can be applied to many different instances (contagion management in social networks). Unfortunately I am unable to leave an educated review since this paper seems to be targeted at a highly technical research sub-community which I have had no prior exposure to.

My primary concern is that this paper might not be accessible to most of Neurips' audience and thus might be better received at more theoretical conferences like STOC or COLT. However, I do not feel as though this should be a disqualifying factor for acceptance, especially if there is a large enough community at Neurips that would find this paper useful.

Going through the theoretical results, the assumptions seem reasonable and I can't see any glaring red flags in their resulting theorems.

I do not feel like I have enough context to critically interpret the significance of the improvements in the experimental results.

Overall this paper seems sound to me, but I have concerns about the accessibility of the paper. Hence I recommend a borderline accept with very low confidence. I think the authors could significantly improve the paper if they spent some time providing more intuition in the first few sections and also added some concrete examples to aid the interpretation of their definitions.

---

> ### Author Response · Authors · 2022-07-30
> **Author response**
>
> We thank the reviewer for their time and constructive comments.
>
> - $\textbf{Neurips’ audience.}$ We agree with the reviewer that some technical contributions of this paper may be better received at other conferences: for example, the proposed algorithms may be of interest to the STOC community, and our analysis of the generalization bounds falls into the central topics of COLT. We wish to submit this work to Neurips because we believe that Neurips is more interdisciplinary than other top venues. For example, social influence has attracted a fair amount of attention from Neurips researchers [a, b]; a few seminal works of data-driven decision making (and inverse engineering) are from Neurips [c, d, e]; our theorems are inspired by existing works that are also published at Neurips [f, g]. We believe that our findings may interest researchers from multiple Neurips communities.
>
> - $\textbf{Presentation.}$ We thank the reviewer for this comment, and we are happy to add more intuitions/examples to make this paper more accessible to Neurips’ audience, by leveraging the extra page of the cameral-ready version. In this regard, it would be very helpful if the reviewer could advise on the definitions that need further clarifications.
>
> - $\textbf{Theoretical assumptions vs. real-world practices.}$ We share the same view with the reviewer that the assumptions of this work only remotely mirror real-world practice, and this is mainly because the details about how real social media are operated are often not available to the public. The two considered tasks, information enhancement and information containment, are designed to abstract strategies for launching advertising campaigns and misinformation prevention. Such applications have been widely studied and implemented by real-world network platforms [g]. Furthermore, from a high-level perspective, we believe that task migration is a future topic in developing data-driven pipelines (for the general case). It is our aspiration to explore the issue of task migrations in more general settings or other applications, and this paper attempts to offer a primitive formal study on a specific application.
>
> [a] Du, Nan, Le Song, Manuel Gomez Rodriguez, and Hongyuan Zha. "Scalable influence estimation in continuous-time diffusion networks." Advances in neural information processing systems 26 (2013).
>
> [b] He, Xinran, Ke Xu, David Kempe, and Yan Liu. "Learning influence functions from incomplete observations." Advances in Neural Information Processing Systems 29 (2016).
>
> [c] Donti, Priya, Brandon Amos, and J. Zico Kolter. "Task-based end-to-end model learning in stochastic optimization." Advances in neural information processing systems 30 (2017).
>
> [d] Wilder, Bryan, Eric Ewing, Bistra Dilkina, and Milind Tambe. "End to end learning and optimization on graphs." Advances in Neural Information Processing Systems 32 (2019).
>
> [e] Dong, Chaosheng, Yiran Chen, and Bo Zeng. "Generalized inverse optimization through online learning." Advances in Neural Information Processing Systems 31 (2018).
>
> [f] Wu, Yuanbin, Man Lan, Shiliang Sun, Qi Zhang, and Xuanjing Huang. "A learning error analysis for structured prediction with approximate inference." Advances In Neural Information Processing Systems 30 (2017).
>
> [g] Rahimi, Ali, and Benjamin Recht. "Weighted sums of random kitchen sinks: Replacing minimization with randomization in learning." Advances in neural information processing systems 21 (2008).
>
> [h] Wingfield, Nick, Mike Isaac, and Katie Benner. "Google and Facebook take aim at fake news sites." The New York Times 11 (2016): 12.

---

> > ### Comment · Reviewer_egd8 · 2022-08-09
> > **Satisfied with response**
> >
> > Dear Authors,
> >
> > Thank you very much for your in depth response I am satisfied with your response and have upped my score as a result.
> > I think the main piece of intuition I'd like to see added would be for the stochastic diffusion model defined in 2.1, and the various tasks defined in 2.2. As someone that was unfamiliar with that area it took me a bit of time to parse what ended up being rather natural definitions for the problem setting in the end.
> > Since they make heavy use of graphs I think visual illustrations of these problems would be most beneficial (e.g. tikz figures).

---

### Review · Ethics_Reviewer_kJmU · 2022-08-01

**Recommendation:**

I would strongly recommend for the authors to think on the implications of their work and compose a detailed ethics statement and an in-depth ethics assessment regarding this piece of research.

Given that social contagion management tasks include both diffusion enhancement and diffusion containment, and can consequently be used either for fighting discrimination and dangerous speech, they can just as easily be used for spreading misinformation and hate speech and causing harms to marginalised groups, if used by bad actors. It is imperative to consider the consequences of releasing generic social contagion management frameworks and making this knowledge open, contextualizing this research given the known ongoing and historical cases of social media being weaponized for propaganda and radicalization.

As it stands, the paper fails to engage with these implications.
The final decision on the publication should take into account these additional justifications, hopefully provided by the authors in the interim.

**Ethical Issues:**

Yes

**Ethics Review:**

The authors propose a way of utilising inverse decision-making with task migrations towards a development of a generic framework for addressing decision-making tasks in social contagion, e.g. diffusion enhancement / diffusion containment. In practical terms, as given by the authors, these types of methods could be applicable to help with scenarios involving fighting misinformation and suppressing violence-promoting messages, as well as facilitate marketing campaigns.

The breadth and significance of such applications make it imperative to carefully assess the impact of this work and to provide an ethics statement that dives into such opportunities and risks.

After all, isn't this dual-use? We can say that it may help fight misinformation, but wouldn't it be just as useful in maliciously spreading misinformation and facilitating state-sponsored propaganda by authoritarian regimes? It's just a question of the intentions of the user, as well as the type of content - the method itself is agnostic to whether the use is good or bad, no?

Dual use does not automatically invalidate this (and similar work), but the very conscious decision to publish and publicise such results at venues where they are likely to be getting a lot of attention - not only from good actors - does have ethical implications.

With that in mind, the authors need to provide a detailed ethics statement, a risk assessment, and consider risk mitigations.

---

### Review · Ethics_Reviewer_r2zb · 2022-08-04

**Recommendation:**

As the authors note in their response, the work should explicitly acknowledge the potential misuses of the proposed work.

**Ethical Issues:**

Yes

**Ethics Review:**

I understand why the paper was flagged for inappropriate potential applications and impact. There is definitely a lot of value of the proposed work grounded on potential positive applications (e.g. curb misinformation), but this a case of potential dual usage in which the same technology could be used, for example, by governments seeking to counter pro-democracy movements, human rights advocates, and other types of community organizing.

---

### Meta-Review · Area_Chair_nAX7 · 2022-08-24

**Recommendation:** Accept
**Confidence:** Less certain

**Metareview:**

Strengths:
* novel formulation for task migration in social management tasks
* theoretical analysis: generalization bound
* results shed light on certain possible design choices
* adequate empirical evaluation on simulated data

Weaknesses:
* formalization may be too restrictive to capture realistic settings (e.g., observing only one type of task)
* connections to some related literature not clearly established
* some concerns regarding baselines used in experiments, or lack thereof
* societal implication not discussed or properly acknowledged in authors’ response **(see ethics section below)**


Summary:
All reviewers agree that the proposed problem of task migration for network diffusion is interesting, and that the proposed formal framework is elegant; some reviewers, however, found the framework to be somewhat restrictive in its ability to relate to real-world diffusive processes. Theoretical results seem sound, but to fully appreciate their novelty, it would help if the authors provide more details and make concrete connections to the literature on which they draw. The authors’ response to reviewers questions were helpful in this regard. Experimental results look encouraging; nonetheless, several reviewers raised concerns regarding the adequacy of certain baselines, or the lack of comparison to methods that include explicit diffusion modeling. Authors’ responses in this regard were only partially satisfactory.


**Ethics:**
Reviews by two designated ethics reviewers strongly suggest that the paper frames the decision task it studies in a very one-sided manner, presenting mostly possible benefits (e.g., minimizing the spread of misinformation) and lacking to acknowledge for its evident risks (e.g., maximizing the spread of misinformation). The dual objectives studied—diffusion enhancement (Problem 1) and containment (Problem 2)—make it very clear that virtually every concrete task in this space has potential dual use. **Unfortunately, the authors do not discuss this in the paper, nor were their responses in the discussion appeasing.** Due to this, acceptance is made conditional on the authors’ ability to clearly convey in their writing, and *in the early parts of the paper*, the risks that naturally follow from their proposed work, as they relate to the societally-significant applications they discuss.



**Award:**

No

---

### Decision · Program_Chairs · 2022-09-14

Accept